# Deriving Information on Play and Playfulness of 3–5-Year-Olds from Short Written Descriptions: Analyzing the Frequency of Usage of Indicators of Playfulness and Their Associations with Maternal Playfulness

**DOI:** 10.3390/bs12100385

**Published:** 2022-10-08

**Authors:** Nancy Tandler, René T. Proyer

**Affiliations:** Department of Psychology, Martin Luther University Halle-Wittenberg, Emil-Abderhalden-Straße 26-27, 06108 Halle (Saale), Germany

**Keywords:** play, playfulness, person perception

## Abstract

Playfulness is an individual differences variable that enables people to (re-)frame almost any situation into an entertaining, amusing, intellectually stimulating and/or personally interesting situation by interacting playfully with others, by resolving tension, by liking complexity over simplicity and/or by having a preference for unusual topics, persons and/or activities. We asked 208 German-speaking mothers of 3–5-year-olds to describe their child in 5–10 sentences. Using a list of criteria for playfulness (e.g., actively initiating humor, playful exchange with others or widespread interests), we found that mothers used, on average, two playful characteristics to describe their child (17% did not report any). Greater usage of playful descriptors in the written texts was positively related mainly to greater other-directed and intellectual playfulness of the mothers. The findings are encouraging and suggest that the list of playful criteria in descriptions of children could be used in the study of inter-individual differences in playfulness in young children.

## 1. Introduction

Play (the actual behavior) and playfulness (the propensity to play; a personality trait) in humans are typically studied among children [1,2,3,4,5,6,7]. Only recent years have seen an increase in research on adult play and playfulness [8,9]; the latter has included testing for familial aggregations [10], effects of assortative mating [11], relationship satisfaction [12] and sexuality [12,13], to name but a few. An understudied question is what *perceptions* parents have about the playfulness of their young children. One way to address this question would be to ask the parents directly. However, conventional wisdom suggests that “all children are playful” or “all children play”, and such implicit theories might affect parental ratings. Hence, we decided on a different approach in this study and asked our participants to write 5–10 sentence descriptions of their child, and we searched for hints about playfulness in these descriptions.

Playful people can (re-)frame almost any situation into an entertaining, amusing, intellectually stimulating and/or personally interesting situation by interacting playfully with others, by resolving tension, by liking complexity over simplicity or by having a preference for unusual topics, persons and/or activities [8]. Early research [2,3,7] identified characteristics of playfulness in young children and there are checklists that can be used to examine inter-individual differences in playfulness in young children. To the best of our knowledge, however, no study has tested whether such characteristics could be used to identify inter-individual differences in playfulness in children. Our study aims to derive and apply a category system that allows information to be gathered on the playfulness of 3–5-year-olds from short written descriptions and to assess whether such information relates to self-ratings of the playfulness of their mothers.

Observed playfulness in young children is frequently related to higher levels of confidence and imagination but also to aggression and mischievousness [7]. Although there are reports that, for example, teachers sometimes think of their playful pupils as being disruptive and that they need more attention than their less playful colleagues, they are also seen as being, among other variables, more alert and creative [7,14,15]. Lieberman [7] proposes that “spontaneity”, “manifest joy” and “sense of humor” are the basic components of playfulness in children. For “sense of humor”, she differentiates among affect (joking, entertaining) and cognition (wit, punning); “spontaneity” unfolds into social, physical and cognitive components, whereas for “manifest joy” she differentiates between laughter, pleasure and preference. Observed playfulness in children was also positively associated with divergent thinking [16]; teachers’ descriptions of young children (age 2–5 years) high in cognitive spontaneity were bright, affectionate, confident, curious, cute and imaginative [7], and also over-excitability [17]. Playfulness in children is also positively related to key aspects of emotional adjustment, such as self-confidence and independence [2], positive affect, the tendency to express oneself [18], adaptive coping strategies [19], self-regulation and happiness [20] and positive play interactions with peers [6]. However, from the last year of kindergarten up to the third grade, teachers perceive playful boys more negatively than their *less* playful counterparts (i.e., as rebellious, intrusive and “class clowns”). After some time, peers also seemed to assimilate this negative teacher impression about playful boys. In contrast, teachers’ and peers’ perceptions of girls varied independently from girls’ playfulness level [4]. Within this research field, there is compelling evidence for five playfulness dimensions that have been frequently assessed in children: physical, social and cognitive spontaneity, as well as manifest joy and sense of humor [2,7].

We expect that such categories can be retrieved in parental descriptions of their children. Such descriptions have already been successfully used to study personality traits in children as young as 3 years old: for example, the Big Five personality traits [21] and character strengths [22]. Usable descriptors of the respective traits are identified and then applied to free descriptions of children. This line of research aims to identify a catalogue of “those individual differences in temperament and personality that [are] important for parents and teachers in ordinary daily life. What individual differences do parents and teachers actually see as most salient?” [23]. We apply these questions to the study of playfulness. In line with Kohnstamm et al. [24], units of analyses can be “an adjective, noun, or phrase referring to a description of [playful] behavior, personality characteristic, or ability” (p. 5) relevant to playfulness. Furthermore, we expect that people mostly write about those characteristics in their child that “they think are most important or basic” (p. 5) and that these characteristics will also carry information about how playful they see their child.

We see the descriptions as non-direct assessments of children’s playfulness using largely spontaneous written descriptions of children. Our study is based on that of Park and Peterson [22], who collected written parental descriptions of morally positively valued traits (i.e., character strengths) of young children. They counted instances of either direct mention of a strength (or synonym) or the description (or usage) of a strength by the child as described by their parent. We conducted a literature search to determine the most frequently mentioned indicators of playfulness in children (as a proxy for those that are most “important”) as a basis for the analyses (see Section 2.3 for details) and screened the descriptions by the mothers for these indicators.

Additionally, we collected self-ratings of the mothers on their own playfulness, with the intention of relating these to scores derived from the written descriptions of the children. Based on findings for familial aggregations of playfulness, we expected that there would be a positive relationship between self-ratings of playfulness in mothers and derived playfulness scores for the children.

## 2. Method

### 2.1. Procedure

We recruited our participants using social media (Facebook/Meta, online discussion sites for parents) and notices in day care centers, pediatrician offices, places for activities frequently pursued by children and in university buildings. Mothers with at least one child aged 3–5 years were invited to take part. We collected the data in an online survey using SoSci Survey [25]. We conducted and designed our research according to the code of good practice in online-based testing [26]. Mothers first provided demographic information about themselves and about one of their children. Then, using an open-ended question format, mothers had to freely describe their child and self-report their own playfulness using questionnaires. Participation was voluntarily and no compensation was provided. Completion of the questionnaires took about 20 min.

### 2.2. Sample

Our sample included 208 German-speaking *mothers* whose age ranged between 21 and 48 years (*M* = 33.1, *SD* = 5.0). Almost three-quarters were married (70.7%), 18.3% were in a long-term relationship and 11.0% reported being single or living in separation from their partners. The mothers spend on average 5.4 hours with the target child (*SD* = 2.2, range = 0–15) during a regular weekday and most of the women were the biological mothers (98.1%). The others were stepmothers (0.5%) or had adopted the child (1.4%). Regarding their highest educational level, 25.5% had completed vocational training, 13.0% reported having an “Abitur” (a school-leaving diploma that qualifies for study at a university), 44.2% held a university degree, 0.5% reported having no degree, 23.1% attended the degree from the lower and medium secondary school tracks and 5.8% did not report their educational level.

Most children were 3 years old (38.5%), 30.8% were age 4 years and 30.8% were age 5 years; 89 (42.8%) were girls and 119 (57.2%) were boys. Most of the children did not suffer from a serious illness or disability (93.3%); almost all children attended a day care facility (99.0%) and more than half of them (54.3%) were engaged in a regular activity related to music, dancing or sports. One-third were only children (37.51%); most of the children had one sibling (40.9%), 17.3% had two siblings and 4.4% had three or four siblings.

### 2.3. Measures

*Mothers’ non-direct/implicit perceptions of their child’s playfulness.* Mothers’ perceptions were measured by combining a qualitative assessment with a quantitative approach. As in the study by Park and Peterson [22], the mothers were instructed to freely describe one child in their own words using 5–10 sentences and try to provide a thorough description of the child. 

These descriptions were then segmented into minimum units of information about all aspects of the children. Phrases not related to any aspects of the children were omitted from further analyses. The remaining information units were coded according to whether they contain (=1) or do not contain (=0) criteria for playfulness. These criteria are based theoretically on previous research into playfulness [2,7,8,27,28,29,30,31] and playfulness aspects that appear to be applicable to children’s behavior (e.g., playful behavior, creativity, lightheartedness and joy of joking around) [31,32,33]. In line with previous research [22,34], information units were not coded if they contained playfulness information in a *negative* way (e.g., “rejects creative activities” or “he needs a regular schedule, otherwise he can get confused”).

The following 15 playfulness criteria were selected from mothers’ free texts: “actively initiating humor”, “playful exchange with others”, “widespread interests”, “active and explorative behavior”, “carelessness”, “risk-taking”, “need for movement”, “urge to talk”, “creativity”, “imagination”, “cognitive spontaneity”, “to do things on his/her own in a playful way”, “enjoying fooling around and being silly”, “own style” and “willpower”. These playfulness criteria with their definitions, coding rules and examples are presented in more detail in Table 1.

For testing the inter-rated agreement, two independently working raters (i.e., specifically trained undergraduate students in psychology who are familiar with playfulness research) coded 30 randomly selected descriptions by mothers. They agreed on the majority of the entries; the inter-rater reliability was *r* = 0.80. Afterwards, the two raters discussed their classifications of the segments they had not agreed upon until agreement was reached. This helped in establishing a joint coding strategy. For the remaining descriptions, only one rater was employed.

Finally, we computed a playfulness score for each child based on the number of playful criteria described by their mothers: the *absolute non-direct playfulness score*.

The average length of the texts based on mothers’ free descriptions was *M* = 79.63 words (*SD* = 49.57). The longest description contained 358 words and the shortest description consisted of just 5 words. The number of absolute playfulness characteristics correlated moderately with the description lengths (*r* = 0.41, *p* < 0.000). Thus, we additionally computed the proportion of the absolute number of playfulness descriptions to the overall text lengths: the *relative non-direct playfulness score*.

*Mothers’ explicit descriptions of their child’s playfulness.* We additionally used a one-item rating scale (“The described child is a playful person”) to directly assess mothers’ perceptions of their child’s playfulness. Mothers had to respond on a seven-point Likert scale ranging from 1 (“not at all”) to 7 (“totally agree”). On average, mothers reported that their children are rather playful (*M* = 5.74; *SD* = 1.18; range = 1–7).

*Mothers’ self-reported playfulness.* We used the adult version of the OLIW questionnaire [8] to assess the four *facets* of playfulness, namely: *Other-directed* (O; e.g., “I can use my playfulness to do something nice for other people, or to cheer them up”), *Lighthearted* (L; e.g., “I don’t worry about most of the things that I have to do, because there will always be some kind of a solution”), *Intellectual* (I; e.g., “In the final account, a discussion is nothing other than playing with and an exchange of ideas”), and *Whimsical* (W; e.g., “I like to swim ‘against the stream’”). The instrument consists of 25 items to which participants respond on a seven-point Likert scale (1 = “not at all” to 7 = “totally agree”). Each facet contains seven items, with the exception of the *Intellectual* facet that only contains three items in our study. We had to eliminate the reverse-coded items due to labeling problems during data collection, as the internal consistency including the reverse scoring was 0.46; and we had to eliminate the item representing “handling of boredom” because we argue that this item assesses workload rather than intellectual playfulness in mothers with young children. In our sample, the internal consistencies for the four playfulness facets were as follows: O = 0.73; L = 0.74; I = 0.64; W = 0.76.

**Table 1 behavsci-12-00385-t001:** Criteria catalog of mothers’ perception of children’s playfulness and its coding scheme, examples and mentioned numbers (*n*) for each playfulness criterion.

Playfulness Characteristics	Literature	Definitions and Coding Rules	Examples	*n*
1. Actively initiating humor	[2,8,28,35]	The quality or expression of being cheerful, joking, funny or curious Ability to perceive or convey humor Expressions of humor: making jokes, cheering others up or entertaining them	“People around him call him a very funny young man”, “he carries everybody along with his jokes”, “child tries to soothe via distracting by playing”	16
2. Playful exchange with others	[2,8,28,35]	Child makes playful contact with other children or adults Certain (favorite) games that require at least one other person as a player	“Enjoys playing with other children”, “her favorite games are roleplays”, “we often find ourselves in roleplays”	57
3. Widespread interests	[2,8,28,32,35]	Increased curiosity and/or naming of at least 3 clearly distinguishable areas of interest	“Keen on everything”, “she can bombard someone with questions“, “inquisitive and eager to learn”	65
4. Active and explorative behavior	[2,8,28,35]	Showing clear exploration behavior Child becomes active and investigates the environment independently	“He has to touch and try out everything”, “she loves exploring everything around her”, “investigates everything to find out how it works”	16
5. Carelessness	[2,8,28,35]	Not looking ahead, carefree behavior Flexibility	“Adventurous”, “he likes to object to what we say”, “objects to rules, does not mind consequences, joy is important”	7
6. Risk-taking	[8]	Courage and willingness to take risks or having a risk appetite	“Brave and unflinching”, “has a high risk-taking propensity”	9
7. Need for movement	[2,27,28,35]	Increased activity, in the form of the need for movement/exercise Designations for the child that indicate an increased level of activity	“He needs to move a lot”, “bundle of energy”, “wild nature”	28
8. Urge to talk	[2,28,35]	Increased speaking activity Babbling away, with interlocutor or reactions by interlocutor not essential	“Enjoys talking a lot”, “very talkative”, “she chatters without intermission”	11
9. Creativity	[2,8,28,35]	Clear “creative”/”creativity” expression or ways of acting that require creativity (changing/shaping things)	“Builds and creates things”, “her strengths are in the creative domains”, “when playing, he gives free rein to his creativity”	25
10. Imagination	[2,8,28,35]	Child is imaginative or exhibits (play) behavior that requires imagination (e.g., making up stories) Daydreams	“Gives free rein to her imagination”, “intense daydreaming episodes during daytime”, “she plays imaginatively”	31
11. Cognitive spontaneity	[2,8,28,35]	Child has mental flexibility, the ability to establish relationships Preference for puzzles/brainteasers	“Eloquent”, “glib”, “verbally dominating elder and other children”	22
12. To do things on their own in a playful way	[2,28,30,35]	Child can also play alone or engage in a game alone Not coded: the child can occupy him/herself well alone Special playing behavior Can be excessive in terms of quality, but also in terms of time	“She is good at entertaining herself for long periods with painting or puzzling”, “she can play on her own in her room very creatively and enduringly”, “he enjoys playing loudly and strikingly”, “never gets tired of playing, always wants to play more, refuses to be put off”	46
13. Enjoying fooling around and being silly	[2,8,28,35]	Child enjoys goofing off and clowning around and exhibits these behaviors often and willingly	“When with others, he likes to act the fool”, “bletherer”, “he enjoys getting up to nonsense from time to time”	9
14. Own style	[2,8,28,35]	A style chosen or demanded by the child itself May refer to certain rather unusual habits/behaviors or to outward appearances	“Different from the other girls in kindergarten, her favorite game is…”, “loves to dress herself up”, “very choosy in everything”	15
15. Willpower	[2,28,35]	Child has a strong will, indications of a pronounced own opinion/self-determination In the sense of playfulness: child clearly knows what he/she wants and tries to enforce it Not coded: expressions of stubbornness, maliciousness/petulant behavior	“She increasingly tries to get her way and is very self-confident at it”, “he speaks his mind and behaves consistently in line with his opinion”	43

*Note. N* = 208 mothers.

Additionally, we used the *Short Measure of Adult Playfulness* (SMAP) [30] to assess *global* playfulness, meaning a frequent display of playful activities and high intensity and easy onset of playful experiences. The scale consists of five items (e.g., “I am a playful person”), answers are given on a seven-point Likert scale (1 = “not at all” to 7 = “totally agree”) and the internal consistency for our sample was high (Cronbach’s alpha = 0.89).

## 3. Results

### 3.1. Mothers’ Perceptions of Child Playfulness: Descriptive Statistics

Although mothers were not explicitly asked to write about playfulness or anything related to play behaviors in general (see Section 2.3), most of the mothers used at least one characteristic that fits our criteria for playfulness (see Table 1). On average, they mentioned two playful characteristics of their child (*M* = 1.90, *SD* = 1.36, range = 0–6: i.e., absolute non-direct playfulness score). The average relative non-direct playfulness score was *M* = 0.03 (*SD* = 0.03, range = 0.00–0.29). Figure 1 displays the distribution of the absolute non-direct playfulness scores. Around 17% (*n* = 35) of the mothers mentioned *no* playfulness criteria, while around one-quarter of the mothers mentioned one (26.6%, *n* = 54), two (23.6%, *n* = 49) or three (22.1%, *n* = 46) playfulness criteria. Less than 10% of the mothers mentioned four playfulness criteria (8.2%, *n* = 17), five mothers (2.4%) mentioned five playfulness criteria and two (1.0%) mothers described six playfulness criteria.

The most frequently mentioned characteristics of playfulness were “widespread interests” (mentioned by *n* = 69 mothers), “playful exchange with others” (*n* = 57) and “to do things on his/her own in a playful way” (*n* = 46), while the least-mentioned playfulness characteristics were “carelessness” (*n* = 7) and both “enjoying fooling around and being silly” and “risk taking” (*n* = 9; Table 1). Overall, these findings support the notion that criteria could be retrieved in the descriptions provided by the mothers.

Table 2 shows that there was a robust positive association between absolute and relative non-direct playfulness scores that predominantly derives from the same origin (i.e., the number of descriptions in the written text; see Section 2.3). Mothers’ absolute amount of playfulness descriptions also varied significantly with mothers’ explicit perceptions of the intensity of their children’s playfulness, which is represented by a one-item playfulness rating scale.

### 3.2. Socioemotional Associations of Mothers’ Perceptions of Child Playfulness

The distribution of mothers’ perception of child playfulness did not vary across the children’s gender or age (see Table 3 for descriptive statistics). Mothers with more than three children provided more indicators of playfulness in their children than those with fewer children [*F*(3, 204) = 3.35, *p* = 0.020]. Post hoc tests (least significant difference, LSD) revealed differences between only-children (*d* = 0.03) and children with one (*d* = 0.03) or two (*d* = 0.03) sibling(s).

Mothers’ age was unrelated to their absolute and relative playfulness descriptions of their children as well as to their response on the playfulness rating scale (Table 3). This means that there was no difference in the number of reported playfulness scores regarding whether the mothers were younger or older. Furthermore, the mothers’ family status was unrelated to all playfulness scores of their children: that is, playfulness descriptions by mothers were equal in single mothers and mothers in relationships. Mothers’ educational level was associated with mothers’ absolute amount of playfulness descriptions of their children [*F*(3, 192) = 3.95, *p* = 0.009]. Post hoc tests (LSD) revealed that mothers with lower educational levels (no degree, lower and medium secondary school tracks) reported significantly fewer playfulness descriptions of their children than mothers holding A-levels (*d* = −0.97), a completed university degree (*d* = −1.0) or with vocational training (*d* = −0.89). The average time per week that a mother spends with her children was unrelated to all playfulness descriptions of the children.

### 3.3. Correlates of Mothers’ Perception of Children’s Playfulness with Mothers’ Self-Reported Playfulness

The means, standard deviations and intercorrelations of mothers’ self-reported playfulness across the two questionnaires (SMAP and OLIW) are shown in Table 4. As in other adult samples (e.g., [8]), there was good convergence (*r* = 0.21–0.60) among them. There was a relationship with mothers’ age (*r* = −0.23–0.01; see online Appendix A) and their relationship status (*r* = −0.23–0.01; Appendix A), therefore we controlled for potential effects in the subsequent analyses. Further information concerning the associations of mothers’ self-reported playfulness measures with socioemotional variables is presented in Appendix A.

We computed bivariate correlations between mothers’ perceptions of their children’s playfulness and mothers’ self-reported playfulness (controlling for mothers’ age and relationship status) and also the multiple squared correlation coefficients between the measures of mothers’ perceptions of children’s playfulness and all OLIW facets (lower part in Table 2). Mothers reported more absolute non-direct playfulness descriptions of their children when their self-reported *Other-directed*, *Intellectual* and *Whimsical* playfulness facets were also higher. The self-reported playfulness facets shared 8.8% variance with the absolute number of playfulness descriptions. In addition, mothers’ self-reported global playfulness (SMAP) was positively related to mothers’ amount of non-direct playfulness descriptions of their children.

However, when relativizing the number of mothers’ absolute descriptions on the length of their written texts, this relative non-direct playfulness measure correlated only with mothers’ self-reported *Lighthearted* and *Whimsical* playfulness and shared about 3.3% variance with the playfulness facets (OLIW).

When assessing mothers’ impressions of their children’s playfulness explicitly on our one-item rating scale, the overlap with mothers’ self-reported playfulness reveals the highest associations. This might be due to common method variance (rating scales only). Mothers who described their children explicitly as more playful also had more self-reported *Other-directed* and *Intellectual* playfulness. The OLIW facets explained 11.0% of mothers’ explicit playfulness descriptions. Furthermore, mothers’ self-reported global playfulness (SMAP) was positively related to mothers’ explicit ratings of their children’s playfulness.

## 4. Discussion

Can we derive information about inter-individual differences in how playful a young child (3–5 years) is perceived from short spontaneous written general descriptions of the child (5–10 sentences)? The tentative answer from this study is *yes*. This is in line with findings from earlier research on character strengths [22] and the Big Five personality traits [23,24]. One of the main results of the current study is the development of a list of criteria that can be used for scoring parental descriptions for playfulness. Future research will show whether this will also be useful as an assessment instrument in its narrow sense.

### 4.1. Information about Playfulness in Maternal Descriptions of the Child

Without having been asked directly, mothers reported, on average, two playful characteristics for their child, in an open text format. Hence, mothers spontaneously reported the playfulness descriptions of their child. However, it should be mentioned that about 17% of the mothers did not report any characteristics of playfulness in the descriptions of their child. Of course, this does not mean that their children do not play and it also does not mean that these mothers do not think that play is important. It simply means that there were descriptions that did not refer to playfulness. Future research will show whether there are children for whom playfulness is not a personality characteristic or whether they are deprived of playtime or opportunities to play. The latter may also partially explain why mothers with three or more children perceive more playfulness in their children. In particular, with the strict social distancing rules during the COVID-19 pandemic, (more) siblings potentially provide more and different opportunities to behave playfully than only-children do. This time perspective (i.e., potential differences in outcomes depending on how strict the social distancing regulations were at the time of data collection) adds to the need for replication of the present findings.

The importance of the results of our preliminary implicit playfulness measure is further supported by the positive association with explicit measures of mothers’ perception of their children’s playfulness. Using a single-item rating scale, we asked mothers directly to report the global playfulness level of their children by using the word “playfulness" explicitly. This positive association, although small, can be taken as initial support for the validity of the indirect measure. We further attempted to validate our playfulness list by studying how well our playfulness characteristics (see Table 1) fit the current playfulness models for children and youth – namely, the Liebermann model [2,7], with the five playfulness dimensions of physical, social and cognitive spontaneity as well as manifest joy and sense of humor, and the OLIW model [8,31], with Other-directed, Lighthearted, Intellectual and Whimsical playfulness facets. First, we assigned our playfulness characteristics to the models (based on theoretical reasoning); the initial results are given in online Appendix A. Then, external raters independently checked how well mothers’ free descriptions fitted each dimension of these two playfulness models (OLIW and Lieberman; detailed analyses are described in Appendix A). The characteristics of our playfulness list based on mothers’ free texts are reflected well in both models (Appendix A). In general we obtained small-to-medium associations, and for three of the OLIW facets (Other-directed, Lighthearted and Whimsical) the playfulness associations within these facets were numerically higher than associations with other playfulness dimensions. The same applied for the Lieberman dimensions (Physical Spontaneity, Social Spontaneity, Cognitive Spontaneity and Sense of Humor). Yet, for all dimensions, relationships with other dimensions were found as well.

One might wonder whether mothers’ perceptions of playfulness in children are associated with personal and situational factors—either of the children and/or the mothers. We did not find differences in mothers’ playfulness perceptions according to the child’s age or gender. This is in line with research on young children’s playfulness observed by teachers or external raters. For example, Pinchover [15] showed that overall observed playfulness in nursery school students (age 3–6 years) did not vary with students’ age, and Barnett [4] detected no gender differences in global playfulness in children at the age of 5–6 years.

Our data suggest that children with three or more siblings were perceived as more playful (in the written descriptions) than children with fewer siblings. This is in line with previous results showing that last-born children were also described as being more playful than first- or only-children [36]. Environmental effects may be important here: children with more siblings might receive more stimulation and animation to engage in playful activities and are thus perceived as being more playful; whether or not they truly *are* more playful cannot be deduced from our data, but this would be an interesting topic for follow-up research. One might argue that there are also other factors to be considered, such as time spent playing with the children by their parents or other persons involved in their upbringing, financial and social resources, living conditions, etc. Contrary to earlier studies [36], mothers’ age was unrelated to their descriptions of how playful their children were. Mothers with lower educational levels reported a smaller number of absolute playfulness descriptions than mothers with higher educational levels. This might partly be explained by the fact that they wrote shorter texts than all other mothers [*F*(3, 192) = 5.71, *p* = 0.001]. Accordingly, there were no differences in the relative frequencies. However, it might be interesting to follow this up in future research and test, qualitatively and quantitatively, whether educational level has an impact on the perceived importance of playfulness for the development of a child.

### 4.2. Role of Mothers’ Playfulness

There is a positive relationship between mothers’ self-reported global playfulness (SMAP) and primarily *Other-directed*, *Whimsical* and *Intellectual* playfulness and their usage of indicators of playfulness when writing about their children. This may provide preliminary support for the notion that greater trait playfulness in mothers may contribute to how they see their children (i.e., more playful), even at an implicit level (as derived from the written texts). This is in line with Shen et al. [10], who found that only mothers’ (not fathers’) trait playfulness was positively related to young adults’ playfulness. Whether this, in turn, might have an impact on their educational practices is unclear. Again, Shen and colleagues found that parental playfulness is related to adaptability in their adult children (i.e., ability to adapt to the current and potentially also future environment in which an individual lives). A longitudinal study would be necessary to test whether playfulness derived from parental descriptions leads to greater adaptability in adolescence and/or adulthood. There is preliminary support for the validity of our list of playful incidents: the one-item rating scale of child playfulness (as rated by the mother) yielded a highly comparable correlational pattern.

Our finding that playful mothers potentially observe more playfulness in their children and perhaps also in their daily environment in general is coherent with the definition of adult playfulness. Adult playfulness is defined as the predisposition to frame (or reframe) everyday situations to provide oneself and others with entertainment, stimulation and/or personally interesting experiences [8]. Does this (re-)framing of everyday situations affect the child? Do mothers with higher levels of playfulness create everyday situations and experiences for their children that are different from those created by mothers reporting lower playfulness levels? One can imagine that playful mothers create more stimulating, exciting and inspiring situations that might have an impact on their child’s development (e.g., their adaptability). This also fits with results from observational studies, where independent raters found students trained by playful nursery school teachers to behave more playfully than students of less-playful teachers [15]. Future research will show whether they actually do create such environments and whether playfulness has a direct and/or indirect impact on parenting. Furthermore, as we have only studied mothers, the role of the fathers warrants further caution – not only in the sense of replication of the present findings but also whether differential effects may be visible [37,38].

Results from the field of interpersonal perception support our results that individual characteristics of the mother, such as playfulness, play a mitigating role in perceiving their children’s playfulness (e.g., Funder’s Realistic Accuracy Model) [39]. Interpersonal perception describes how people judge other people’s personality [40]. Research in interpersonal perception demonstrated, for instance, that people who tend to judge others more positively were described by observers as agreeable, content with life, consistent and not hostile, power-oriented or anxious [41], and that people who self-reported more personality disorder symptoms describe others less favorably [42].

Furthermore, the field of interpersonal perception states that perceptions of others are relevant in everyday life because they impact interpersonal behavior [39,43]. Transferred to our results this means, in more detail, that when judging their child’s playfulness, mothers might make a prediction of their child’s future behavior, act accordingly and thus evoke corresponding reactions in the child. This mechanism has already received strong support in the field of self-fulfilling prophecies [44] and must be considered when discussing practical implications.

We operationalized mothers’ playfulness perception by creating an absolute score, reflecting the number of playfulness criteria mentioned in the texts according to our criteria list, and also a relative score (relativized to the text lengths). The associational patterns with mothers’ self-reported playfulness are different for the two scores. Associations of the relative score are scarce (e.g., lighthearted playfulness) and do not resemble those of the absolute score, whereas the absolute score and the ratings scale have comparable associations (e.g., other-directed, intellectual and overall playfulness). We cannot derive practical recommendations from this initial study because the findings warrant replication and extension (e.g., using designs that allow for inferring causality or take multiple measurement times into account). However, the data allow for speculation about the effects of interventions to help parents foster playfulness in their children, taking the data from the absolute scores into account rather than the other scores. First, the absolute number of mothers’ perceptions is the “thin slice of behavior” [45,46] of interest, in the sense that it is the original form in which mothers expressed their perceptions. In addition, for teachers, research has shown that positive expectations are associated with expressiveness [47,48], and it may make a big difference for children whether mothers express their positive or negative perceptions only rarely or extremely frequently. Using absolute numbers of statements allows us to capture this expressiveness, whereas proportional scores would probably obscure it. For example, if mother A expresses one positive and one negative perception of a child and mother B expresses 10 positive and 10 negative perceptions, this would result in the same proportional scores for both mothers, whereas the absolute score would reflect the fact that mother A tends to express their positive (and negative) perceptions a lot.

### 4.3. Limitations

Our cross-sectional design does not allow any conclusions to be made about the direction of causation; that is, our study does not provide answers for whether mothers’ self-reported playfulness increases mothers’ perception of the playfulness of their children, or whether mothers’ perceptions of the children’s playfulness lead to higher playfulness traits in the mothers. Future research should consider longitudinal designs to learn more about the direction of causation and whether implications considering mothers’ level of playfulness are suitable. Of course, an extension to fathers will be needed; for the present research, the decision to study mothers was based on prior research [10] but it is undisputed that fathers also play an important role in children’s development. Furthermore, we did not study same-gender parents. Such parents might be a particularly important group to study because they probably face different environmental demands (such as having to deal with homophobic comments when with their children or answering questions from other children at kindergarten or school). Our sample size is suitable for answering the research questions, but one might argue that it is below what could be recommended based on simulation studies of when correlation coefficients stabilize [49].

This is the first usage of our list of playful behavior indicators in children. We have derived the list from the literature and the indicators cover what is typically mentioned in the literature. However, this may also be a cause for concern. For example, Proyer and Jehle [50] conducted a joint factor analysis of 17 playfulness scales. In short, they found that existing measures lack distinctiveness (e.g., confuse humor and playfulness) [8] and are biased towards extraversion. In addition, influential research such as Lieberman’s [7] model of playfulness in young children may have limitations (e.g., when seeing “manifest joy” as an affective state that is core to playfulness) and alternative measurement approaches exist [6,51]. In short, the distinctiveness of our list and its focus on what is core to playfulness can be improved in future research.

### 4.4. Conclusions

Given the overall positive associations of young children’s playfulness with developmentally important features such as emotional adjustment, adaptive coping, divergent thinking and curiosity, confidence and affectionateness [2,7,16,19], we have demonstrated that the playfulness criteria present in the literature are also perceived by mothers in their young children. Future research should study whether these playfulness criteria are also suitable for predicting children’s behavior. Young children’s playfulness is prominent in the natural language of mothers, which might allow us to take this result as an implicit indicator that children’s playfulness is important in mothers’ perceptions, and that mothers’ observed playfulness can have real-life implications [43]. Previous research demonstrated that playfulness is well observed by others, either close partners [33] or acquaintances [52].

In addition, we have discussed the role of mothers’ playfulness in their perceptions of their child and for the resulting mother–child interactions. This raises the question of whether playfulness interventions for mothers might benefit the children. A recent placebo-controlled online intervention study has shown that short (typically setting aside 10 minutes each evening for seven consecutive days) activities designed to foster playfulness result in increased (self-rated) individual levels of playfulness for up to three months after completion of the activity [53]. Whether such activities could be used to make further developments to facilitate more playfulness in parenting is not yet understood. However, developing a set of such activities and tailoring them to individual levels of playfulness (e.g., fostering other-directed vs. whimsical types of playfulness) in a personalized program could be a fruitful field of research in the future. Providing an educational context that encourages a child’s playfulness often “pays off”, whereas when providing less stimulating environments, caretakers and children sometimes have to “pay a price” in terms of less emotional adjustment and less innovative approaches when encountering complex, ambiguous, unknown or insolvable situations that are part of life.

## Figures and Tables

**Figure 1 behavsci-12-00385-f001:**
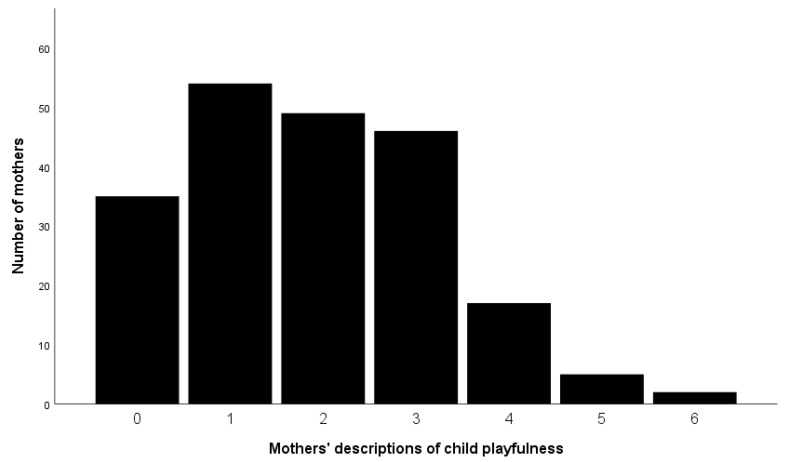
Absolute numbers of mothers’ descriptions of the playfulness of their child (*N* = 208).

**Table 2 behavsci-12-00385-t002:** Intercorrelations of mothers’ perceptions of their children’s playfulness and their correlations with mothers‘ self-reported playfulness while controlling for mothers’ age and relationship status.

*Mothers’ Perception of Children’s Playfulness*			
	Absolute Playfulness	Relative Playfulness	Rating Scale
Absolute playfulness			
Relative playfulness	0.52 ***		
One-item scale	0.15 *	0.05	
*Mothers’ self-reported playfulness*			
Other-directed	0.26 ***	0.07	0.33 ***
Lighthearted	0.08	0.17 **	0.08
Intellectual	0.28 ***	0.07	0.18 *
Whimsical	0.14 *	0.15 *	0.13
*R*^2^/Δ*R*^2^	0.01/0.09	0.01/0.03	0.00/0.11
SMAP	0.16 *	0.01	0.29 ***

*Note. N* = 208 mothers. SMAP, Short Measure of Adult Playfulness.* *p* < 0.05; ** *p* < 0.01; *** *p* < 0.001.

**Table 3 behavsci-12-00385-t003:** Distributions and statistics of mothers’ perceptions of their children’s playfulness across socioemotional features of the children and mothers.

		Mothers’ Playfulness Perceptions of Their Children		
		Absolute Description	Relative Description	One-Item Scale Response
**Sociodemographic Features**		***M* (*SD*)**	***M* (*SD*)**	***M* (*SD*)**
*Child’s gender*				
Male		1.65 (1.28)	0.03 (0.03)	5.63 (1.39)
Female		1.97 (1.40)	0.03 (0.02)	5.82 (0.99)
	*t*(*df*), *p*	−1.67 (206), 0.10	−0.23 (206), 0.82	−1.18 (203), 0.24
*Child’s age*				
3		1.89 (1.32)	0.03 (0.02)	5.68 (1.20)
4		1.84 (1.43)	0.03 (0.04)	5.86 (1.11)
5		1.75 (1.33)	0.02 (0.02)	5.69 (1.24)
	*F*(*df*), *p*	0.19 (2, 205), 0.83	0.66 (2, 205), 0.52	0.48 (2, 202), 0.62
*Child’s siblings*				
0		1.78 (1.34)	0.03 (0.02)	5.84 (1.12)
1		1.81 (1.26)	0.03 (0.02)	5.67 (1.24)
2		1.89 (1.51)	0.03 (0.02)	5.74 (1.01)
3-4		2.22 (1.86)	0.06 (0.09) ^a^	5.44 (1.74)
	*F*(*df*), *p*	0.31 (3, 204), 0.82	**3.35 (3, 204), 0.02**	0.47 (3, 201), 0.70
*Mothers’ age*				
	*r*, *p*	0.03, 0.64	0.02, 0.71	−0.03, 0.70
*Mothers’ family status*				
Single/separated		1.39 (1.37)	0.02 (0.03)	5.78 (1.04)
Relationship/married		1.89 (1.34)	0.03 (0.03)	5.73 (1.20)
	*t(df)*, *p*	−1.66 (206), 0.10	−1.25 (206), 0.21	0.20 (203), 0.83
*Mother’s educational level*				
No degree/lower school track		0.96 (1.00) ^b^	0.02 (0.03)	5.54 (1.22)
Vocational training		1.93 (1.07)	0.03 (0.02)	5.48 (1.37)
A-levels		1.96 (1.38)	0.03 (0.02)	5.82 (1.17)
University degree		1.85 (1.35)	0.03 (0.04)	5.90 (0.99)
	*F*(*df*), *p*	**3.95 (3, 192), 0.009**	0.54 (3, 192), 0.656	1.11 (3, 189), 0.35
*Mothers’ time spent with child*				
	*r*, *p*	0.01, 0.88	0.06, 0.39	−0.11, 0.12

*Note*. *N* = 208 mothers. ^a^ Post hoc tests (LSD) revealed significantly higher scores for groups of 3–4 siblings compared to all other numbers of siblings. ^b^ Post hoc tests (LSD) revealed significantly fewer descriptions by mothers with no degree or lower school track compared to mothers with higher educational levels. Statistically significant findings appear in bold.

**Table 4 behavsci-12-00385-t004:** Means, standard deviations and intercorrelations between mothers’ self-reported playfulness and their age.

Variable	*M (SD)*	O	L	I	W	SMAP
O	4.60 (0.98)					
L	3.91 (0.97)	0.30 ***				
I	3.53 (1.07)	0.54 ***	0.34 ***			
W	3.87 (0.96)	0.32 ***	0.31 ***	0.33 ***		
SMAP	3.92 (1.23)	0.60 ***	0.21 **	0.38 ***	0.19 **	
Age	33.12 (4.99)	−0.20 **	−0.03	0.01	−0.15 *	−0.23 **

*Note. N* = 208. Age in years. SMAP (Short Measure of Adult Playfulness) = global playfulness; O = Other-directed; L = Lighthearted; I = Intellectual; W = Whimsical. * *p* < 0.05; ** *p* < 0.01; *** *p* < 0.001.

## Data Availability

All data are available at https://osf.io/euwkp/ (accessed on 8 October 2021).

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
