# Peer review of "Deriving Information on Play and Playfulness of 3–5-Year-Olds from Short Written Descriptions: Analyzing the Frequency of Usage of Indicators of Playfulness and Their Associations with Maternal Playfulness"

_behavsci, 2022, doi:10.3390/bs12100385_

Round 1
Reviewer 1 Report
The article is well written and articulate in its treatment of the chosen topic.Below are some suggestions for improving the text: It would be useful to specify the nationality of the subjects participating in the research and/or the language used in the questionnaires; On p. 4, line 182 the authors say: 'Two independent raters selected the relevant student descriptions and ...' who are the students being talked about?; On p.8 in Table 2 there isn't correspondence between the contents of the table itself and the note below; On p.11 line 326 there is missing a parenthesis after Tandler, 2020
Author Response
Reviewer #1
The article is well written and articulate in its treatment of the chosen topic
Our response: We are grateful for the positive comment about our work. We are also thankful for the advice that the editing of English language and style need to be edited. We will definitely undergo a professional language editing but we decided to do so after the next revision round to be sure that the final version of our manuscript has adequate language style.
Below are some suggestions for improving the text:
It would be useful to specify the nationality of the subjects participating in the research and/or the language used in the questionnaires
Our response: All questionnaires were administered in German. We added that we have tested German-speaking mothers to the abstract (l. 14/15, p. 1) and also added this information to the description of the sample (l. 118, p. 3).
On p. 4, line 182 the authors say: 'Two independent raters selected the relevant student descriptions and ...' who are the students being talked about?;
Our response: Upon reading the sentence again, we found that the formulation was misleading and we have rephrased the full paragraph to increase its clarity. The revised section (starting on p. 4, l. 184) now reads:
“For testing the inter-rated agreement, two independently working raters (i.e., specifically trained undergraduate students in psychology familiar with playfulness research) coded n = 30 randomly selected descriptions by mothers. They agreed on the majority of the entries; the interrater reliability was r = .80. Afterwards, the two raters discussed their classifications of the segments that they had not agreed upon until an agreement was reached. This helped establishing a joint coding strategy. For the remaining descriptions only one rater was employed.”
Thanks for highlightening this!
On p.8 in Table 2 there isn’t correspondence between the contents of the table itself and the note below
Our response: We have revised the table notes.
On p.11 line 326 there is missing a parenthesis after Tandler, 2020
Our response: We’ve added the missing parenthesis.
Reviewer 2 Report
The study is interesting and contributes to literature and to the filed of early childhood. However, the article writing should be improved so that it will be more accessible and clear to readers:
Background:
1. The background is not clear enough. There is reference to the research question in several different places throughout the literature review, which is confusing. The research questions and hypothesis should be stated clearly at the end of the literature review.
2. More recent work on children's playfulness should be cited.
3. More information about the use of parental descriptions of children's traits, and its importance would be helpful to understand the significance of the study.
Method:
1. The order of the measures that were used in the study is not clear to me. I would start with the Mothers’ non-direct/implicit perceptions of the child playfulness, which is in the center of the study.
2. Line 177-182 should not be in a separate paragraph.
3. I'm not sure about the use of "Mothers‘ explicit description of child’s playfulness". Why did the authors not use a full parental questionnaire to examine children's playfulness? Had they done so, they would have been able to test the reliability of their new research method more thoroughly.
Results:
1. This headline is not clear enough : "3.2. Testing Correlates of Mothers’ Perception of Children’s Playfulness"
2. in general, the result section could be written in a clearer way.
Discussion:
1. I do not find the first paragraph of the discussion accurate, since, as stated before, no comparison was made to existing and reliable measure children's playfulness.
2.The discussion is too long, and can be edited to be more focused and to the point.
Author Response
Reviewer #2
The study is interesting and contributes to literature and to the field of early childhood. However, the article writing should be improved so that it will be more accessible and clear to readers:
Our response: We’re grateful for the, overall, positive feedback and the suggestions provided to further improve the manuscript.
Background:
- The background is not clear enough. There is reference to the research question in several different places throughout the literature review, which is confusing. The research questions and hypothesis should be stated clearly at the end of the literature review.
Our response: We have revised the introductory section fully and have strived to improve the clarity in the writing. Upon reading this section again, we can see the reviewers’ point and hope that the revision has now gained in clarity.
- More recent work on children's playfulness should be cited.
Our response: We have added some more current references.
- More information about the use of parental descriptions of children's traits, and its importance would be helpful to understand the significance of the study.
Our response: We have elaborated on this and have added information about earlier research in the field (on the Big Five personality traits); starting from line 79 on p. 2.
Method:
- The order of the measures that were used in the study is not clear to me. I would start with the Mothers’ non-direct/implicit perceptions of the child playfulness, which is in the center of the study.
Our response: This is a very good point. We reordered our measures according to the reviewer’s suggestion and agree that it enhance readability. Thanks for pointing this out. We begin with the two measures covering mothers’ measures about child playfulness. Then we report mothers’ self-reported playfulness.
- Line 177-182 should not be in a separate paragraph.
Our response: Thanks for pointing this out. We added this paragraph to the former paragraph starting with “These descriptions were then segmented…” that describes our applied coding system.
- I'm not sure about the use of "Mothers‘ explicit description of child’s playfulness". Why did the authors not use a full parental questionnaire to examine children's playfulness? Had they done so, they would have been able to test the reliability of their new research method more thoroughly.
Our response: We agree with the reviewer that the full set of parental descriptions would have been better. In our pre-studies we saw that the willingness by fathers to complete the questionnaire and write about their children was considerably lower than from the mothers. It was a pragmatic decision to focus on maternal ratings. We acknowledge the problem of having only ratings by mothers in the discussion section/limitations.
The reviewer, however, has a good point: It would be interesting to collect both, descriptions by fathers and mothers and then see whether we arrive at different frequencies or even different categories. Also, it might be interesting to have them work on a joint description—or make interviews with them about their child.
Results:
- This headline is not clear enough : "3.2. Testing Correlates of Mothers’ Perception of Children’s Playfulness"
Our response: Thank you for making us aware that this point needed further clarification. We improved the wording of this headline to be more in line with the wording of the related hypotheses (b) mentioned in the background section. The adjusted headline goes as follows: “3.3. Correlates of Mothers’ Perception of Children’s Playfulness with Mothers’ self-reported playfulness“
- in general, the result section could be written in a clearer way.
Our response: We have reorganized the Result section and hope that the revised version is more clearly written and enhances readability.
Discussion:
- I do not find the first paragraph of the discussion accurate, since, as stated before, no comparison was made to existing and reliable measure children's playfulness.
Our response: We agree that there was room for improvement and added more information about comparable findings in personality traits. We included: “This is in line with findings from earlier research on character strengths (Park & Peterson, 2006) and the Big Five personality traits (Kohnstamm et al., 1994, 1998).”
- The discussion is too long, and can be edited to be more focused and to the point.
Our response: Thank you for making us aware that this point needed further clarification. We have reordered some text segments and shorten others in order to reduce on the lengths of the discussion and to avoid redundancies.
For example, we reduced the definition of adult playfulness in section 4.2. to avoid overlaps, we reordered findings on playfulness interventions in the conclusion section and we also reordered section 4.2. covering the role of mothers’ playfulness.